# Effects of Apnea-Induced Hypoxia on Hypoalgesia in Healthy Subjects

**DOI:** 10.3390/sports12110294

**Published:** 2024-10-28

**Authors:** Cristian Mendoza-Arranz, Omar López-Rebenaque, Carlos Donato Cabrera-López, Alejandro López-Mejías, José Fierro-Marrero, Francisco DeAsís-Fernández

**Affiliations:** 1Research Group Breatherapy, Centro Superior de Estudios Universitarios La Salle, Universidad Autónoma de Madrid, 28023 Madrid, Spain; 201011149@campuslasalle.es (C.M.-A.); deasisfernandez@yahoo.com (F.D.-F.); 2Departamento de Fisioterapia, Centro Superior de Estudios Universitarios La Salle, Universidad Autónoma de Madrid, 28023 Madrid, Spain; omarlopezrebenaque@gmail.com (O.L.-R.); carloscabrerafisio@gmail.com (C.D.C.-L.); 3Department of Physiotherapy, Faculty of Health Sciences, Universidad Europea de Canarias, 38300 Santa Cruz de Tenerife, Spain; alelopezfisio@gmail.com; 4Motion in Brains Research Group, Centro Superior de Estudios Universitarios La Salle, Universidad Autónoma de Madrid, 28023 Madrid, Spain

**Keywords:** breath-holding, hypoventilation, pain sensitivity, hypoalgesia

## Abstract

Introduction: Exercise-induced hypoalgesia is a phenomenon in which exercise bouts induce a reduction in pain sensitivity. Apnea training involves similar characteristics that could potentially induce hypoalgesia. Objectives: The objectives of this study are to explore the effect of apnea training on hypoalgesia; assess the correlation between conditioned pain modulation (CPM) response and apnea-induced hypoalgesia; and examine the association between hypoalgesia with hypoxemia, and heart rate (HR) during apnea. Methods: A randomized controlled trial was conducted comparing a walking protocol employing intermittent apnea compared with normal breathing in healthy volunteers. Hypoalgesia was tested with pressure pain thresholds (PPTs) and CPM. Oxygen saturation (SpO_2_) and HR were also tested. Results: Relevant but not significant changes were detected in the thumb (MD = 0.678 kg/cm^2^), and tibialis (MD = 0.718 kg/cm^2^) in favor of the apnea group. No significant differences were detected in CPM. The apnea group presented lower SpO_2_, but HR values similar to those of the control group during the intervention. Basal CPM and intrasession hypoxemia significantly correlated with the PPT response. However, HR did not correlate with the PPT response. Conclusions: The current results suggest a trend, though not statistically significant, toward an improvement in the PPT in favor of apnea training compared to normal breathing. Nevertheless, subjects who presented greater basal CPM and lower oxygen saturation during the session presented a greater PPT response, suggesting the possibility of mediators of response. Future investigations should clarify this phenomenon.

## 1. Introduction

Exercise-induced hypoalgesia (EIH) is a phenomenon in which exercise induces a reduction in pain sensitivity, first described by Black et al. [1]. Researchers currently employ a wide range of methods for assessing EIH [2]. A recent meta-analysis revealed that a single session of aerobic or resistance training can induce hypoalgesia in healthy individuals [3].

EIH appears to be produced systemically, not only occurring in the body locations trained, but also in remote body regions [4]. Moreover, diverse exercise intensities elicit EIH, wherein pain tolerance, the heat pain threshold, and, somewhat disputedly, the pressure pain threshold (PPT) appear to be augmented by higher exercise intensities [5].

Other modalities can produce EIH, such as blood-flow restriction training [4]. This effect is partially mediated by the opioid system immediately after the training session, whereas the endocannabinoid system does not appear to contribute; however, the involvement of the opioid system appears to be negligible or absent for 24 h following the training session [6].

It has been observed that ventilatory maneuvers such as breath-holding can induce pain sensitivity changes [7]. Several hypotheses stand the role of baroreflex in this response [8]. However, breath-holding hypoalgesia could also be derived from similar action mechanisms related to exercise, as widely studied in the literature.

There is scarce evidence concerning the potential hypoalgesic effects of apnea. We postulate that apnea training might induce hypoalgesia through mechanisms such as oxygen desaturation, and changes in heart rate. A randomized controlled trial (RCT) was conducted to explore the impact of low pulmonary volume apnea bouts on hypoalgesia in healthy subjects.

## 2. Materials and Methods

We conducted a parallel RCT with 1:1 allocation. This trial followed the Committee on Publication Ethics (COPE) guidelines and was approved by La Salle Campus Universitario Ethics Committee, Madrid, Spain (Ref CSEULS-PI-029/2022). Protocol registration was performed in Clinical Trials Gov (NCT05991141).

### 2.1. Participants

A non-probabilistic sampling method was implemented in CSEU La Salle and its surroundings (Madrid, Spain). Physical and digital adverts were employed to reach our target participants. To ensure homogeneity in the PPT, CPM, HR, and oxygen saturation (SpO_2_) characteristics within the sample, the following inclusion criteria were employed: (1) age range of 18 to 30 years [9]; (2) absence of current pain symptoms; and (3) SpO_2_ levels ≥ 95%. The following exclusion criteria were also applied with the same aim: (1) recurrent pain during last month; (2) pharmacological treatment; (3) drug consumption; (4) self-harming behaviors [10]; (5) pregnancy or potential pregnancy; (6) splenectomy or spleen disease [11]; (7) engagement in moderate- or high-intensity physical activity within 24 h before the study [12]; (8) absence of sleep the previous night [13]; and (9) alcohol intake 24 h before the study [14]. The participants were included if they met the selection criteria and after providing their consent for participation. Due to the exploratory purposes of this study, a small sample of 30 subjects was included. The participants provided informed consent, being informed about the potential risks of breath-holding maneuvers such as dizziness and loss of motor control, among others. However, to preserve security within the trial, the training session would be stopped immediately if any of these signs were observed.

### 2.2. Randomization and Blinding

The sex demographic has proven to be a confounding factor in the PPT [15] and it might also be in CPM [16] basal values. We performed a cluster randomization by sex to ensure an equitable distribution across groups. The randomized allocation process was conducted by an external researcher before the start of this study, employing GraphPad Quickcalcs–Random Number Calculators (GraphPad Software Inc., La Jolla, CA, USA). The evaluator who assessed the PPTs and CPM remained blinded to the participants’ group allocation, as well as to the data pertaining to SpO_2_ and HR.

### 2.3. Interventions

In both the experimental and control groups, participants engaged in treadmill walking at a velocity of 5.5 km/h with an inclination of 5° for 6 min. The control group (CG) followed this protocol while maintaining normal breathing. Conversely, the apnea group (AG) incorporated a cyclic breathing pattern during the intervention, alternating between 5 s of normal breathing and 10 s of low-volume apnea. This cycle of normal breathing followed by apnea was repeated a total of 24 times (6 min).

In previous studies, different interventions employed the intervallic apnea protocols with durations of 6 s–24 s [17], 40 m–30 s [18], 25 m–30 s [19], and 15 m–30 s [20]. However, these protocols were conducted with recreational athletes. In the present study, which was performed on untrained individuals, the apnea duration was reduced to ensure compliance with the intervention protocol, and the breathing recovery periods were adjusted accordingly. A recent systematic review concluded that short, repeated apneas (6 to 30 s) appear to be the most effective method for producing a hypoxic stimulus in those unfamiliar with apnea training [21].

To perform this respiratory technique, immediately after each inspiration, the participants were asked to perform a passive expiration and then hold their breath 10 s until the next inspiration. According to previous studies [22], this expiration maneuver reduces SpO_2_. Additionally, exercise with apnea bouts also induces increased arterial carbon dioxide concentration, leading to respiratory acidosis [23]. Forced hyperventilation during the 5 s period of normal breathing was prohibited so as not to bias the apnea effects.

### 2.4. Basal Measures

Sociodemographic variables were collected, such as age, sex, and body mass index. Physical activity level, sleep quality, and perceived stress were also assessed, given that they could potentially modify the CPM response, with authors employing the Global Physical Activity Questionnaire (GPAQ), Pittsburgh Sleep Quality Index (PSQI), and Perceived Stress Scale, respectively. These tools were administered in their Spanish-validated versions [24,25,26]. Basal SpO_2_ and basal %HR_max_ were analyzed, with the participant in supine position for 2 min. Both variables were measured with a pulse-oximeter (Nonin^®^ Model 9847, Nonin Medical, Inc., Plymouth, MN, USA).

### 2.5. Hypoalgesia Testing

The PPT refers to the minimal amount of pressure required to elicit the initial perception of pain. We employed a pressure algometer (PAIN TEST™ FPX 25, Wagner Instruments, Greenwich, CT, USA) with a round rubber disc of 1 cm^2^, displaying values in kg/cm^2^. We assessed the PPTs on the dorsal base of the distal phalanx of the thumb and the tibialis anterior muscle belly, both in the dominant limb, to explore systemic (upper and lower limb) hypoalgesia. These regions were marked with a pen to repeat the trials in the same place. Three trials were conducted at each location, with 30 s rest periods between trials. The pressure ramp administered was 0.5 kg/cm^2^/s ± 0.1 kg/cm^2^/s, controlled by a metronome. This ramp protocol presents high interrater reliability (intraclass correlation coefficient [ICC]: 0.91) [27]. Similar ramp protocols have shown excellent test–retest reliability (ICC: 0.85–0.99) [28]. The mean value from the 3 trials was employed for statistical analysis.

The CPM is defined as the phenomenon through which the conditioning stimulus affects the test stimulus [29]. Ischemic pain was applied as a conditioning stimulus on the non-dominant arm and the PPT as a test stimulus on the dominant arm. Ischemic pain was applied with a sphygmomanometer at the proximal region of the non-dominant arm. When inflated to 180–200 mm Hg, the participants were asked to actively move their fingers and wrist. The combination of the pressure tests as a test stimulus and the ischemic pain as a conditioning stimulus presented a moderate test–retest reliability for testing CPM in healthy individuals (ICC: 0.64) [30]. The participants notified researchers when the pain intensity reached 7 out of 10 on the Numerical Pain Rating Scale. High-intensity conditioning stimulus exacerbated the CPM response compared to mild intensities, where controversial results appeared [31].

At that point, 3 PPT trials were administered to the dorsal base of the distal phalanx of the thumb in the dominant arm while maintaining the perception of ischemic pain. The mean of the 3 trials was used in the statistical analysis.

Both PPT and CPM assessments were conducted before and immediately following the intervention, with participants in a supine position.

### 2.6. Cardiorespiratory Assessment

HR and SpO_2_ values were videotaped throughout the 6 min interventions and the subsequent 2 min period on a per-second basis to investigate changes during and after the interventions.

From the 6 min interventions, the peak %HR_max_, the minimum SpO_2_, and the means of both variables were calculated and extracted. Time in the 5 zones of %HR_max_ and in the 4 zones of exercise-induced hypoxemia was also explored [32,33]. Theoretical HR_max_ was calculated with the formula “208 − (0.7 × Age)” [34].

### 2.7. Statistical Analysis

The Shapiro–Wilk test was employed to examine normal data distribution between groups for all variables. According to its results, a *t*-test for independent samples or a Mann–Whitney U test was employed to examine differences between. A chi-squared test was employed to find differences between groups in categorical variables.

A mixed ANOVA was employed to test time, group, and time × group interaction effects on the PPTs for the 2 locations. A mixed ANOVA was also employed to evaluate CPM responses for the upper limbs, analyzing basal and conditioned PPT before and after the interventions. Time, group, and time × group interaction effects were explored. Bonferroni post hoc analysis was employed for reporting post-intervention mean differences results in the PPT.

According to normal distribution analysis, Spearman’s rank or Pearson’s tests were used to evaluate linear correlations between the PPT post-pre-intervention changes.

All analyses were performed with Jamovi Software version 2.3.21, with a confidence of 95%, considering statistical significance when *p* < 0.05.

## 3. Results

### 3.1. Participants Sample Description

A total of 30 healthy volunteers (aged 19–29 years) completed all procedures. Baseline values showed no significant differences, except for SpO_2_ basal value (*p* = 0.035). However, it should be noted that this difference was not considered clinically relevant given that all participants in both groups exhibited SpO_2_ levels ≥ 95%, and the mean difference between the groups was only 1%, not being considered of relevance on the overall response of participants to the interventions. Demographic and baseline data are displayed in Table 1.

### 3.2. Pressure Pain Threshold

The mixed ANOVAs showed no significant effects in the PPTs either in the thumb or tibialis muscle. Nevertheless, the AG tended to present a greater PPT response in both locations (see Table 2 for mixed ANOVA results).

Post hoc tests revealed a non-significant difference in the post-intervention PPT values in the thumb (MD [Exp-Cont] = 0.678 kg/cm^2^, SE = 0.441, t = 1.537, *p* = 0.813), or tibialis (MD [Exp-Cont] = 0.718 kg/cm^2^, SE = 0.706, t = 1.018, *p* = 1).

### 3.3. Conditioned Pain Modulation

The mixed ANOVA revealed only a significant time effect (*p* = 0.004) (see Table 2 for mixed ANOVA results).

The correlation analysis indicated a significant positive association in the AG between the basal CPM response (conditioned PPT − basal PPT: 0.75 ± 0.86) and PPT change (post PPT − pre PPT) in the thumb (thumb PPT change: 0.41 ± 0.78; rho = 0.729, *p* < 0.001) and tibialis muscle (tibialis PPT change: 0.41 ± 0.80; rho = 0.550, *p* = 0.002).

### 3.4. Heart Rate Response

Participants in both groups only displayed %HR_max_ values in zones 1–3, given that they did not reach zones 4 and 5. No differences were observed regarding the time expended in HR zones between groups (*p* > 0.05) (see Figure 1 and Table 3 for mean %HR_max_ values displayed during the training session).

### 3.5. Oxygen Saturation Response

The AG displayed SpO_2_ values from normoxemia to severe hypoxemia zones, whereas the CG only reached the moderate hypoxemia zone (see Figure 2). The CG spent more time than the AG in the normoxemia zone (*p* = 0.002). The AG spent more time than the CG in the moderate and severe hypoxemia zones (*p* < 0.001) (see Table 3).

In the AG, the minimum SpO_2_ values significantly correlated with the PPT change in the thumb (r = −0.551, *p* = 0.033), indicating higher PPT responses in those who reached lower minimum SpO_2_ values (see Figure 3). In addition, time in the mild hypoxemia zone negatively and significantly correlated with the PPT change in the thumb (r = −0.673; *p* = 0.006). This information supports an association in which shorter times in the mild hypoxemia zone correlate with higher PPT changes in the thumb. No other correlations were identified between SpO_2_ values and PPT changes.

## 4. Discussion

### 4.1. Main Findings

The main finding of the present study indicates no statistically significant difference in the PPT between apnea and normal breathing. Although the AG showed relevant PPT mean differences (0.678 kg/cm^2^ in thumb and 0.718 kg/cm^2^ in tibialis), these differences did not reach significance, potentially due to methodological constraints such as the small sample size in this exploratory trial.

Additionally, a significant positive correlation was observed between the CPM response and changes in the PPT observed for both the thumb and tibialis muscle in AG. This finding suggests that individuals with stronger CPM responses may experience a more pronounced hypoalgesic effect following apnea training.

### 4.2. Cardiovascular and Respiratory Responses

This study also compared %HR_max_ and SpO_2_ between groups. As expected, AG participants showed significantly lower SpO_2_ levels, and these values were significantly associated with changes in the PPT. This suggests that the degree of hypoxemia reached during apnea training may be a key mediator of changes in pain sensitivity. Future interventions should explore the mediating role of hypoxemia on induced hypoalgesia, as several factors such as minimum saturation reached or exposure duration to lower saturation levels could be of interest for targeting interventions.

On the other hand, no significant differences were found in %HR_max_ between groups. Previous research found that HR decreased during [35] and increased immediately after the apnea bout as a compensatory mechanism derived from hypoxemia [36]. In our study, the AG performed low-volume intermittent apnea bouts while walking, displaying similar %HR_max_ values as breathing normally when walking. We hypothesize that intermittent apnea bouts (10 s) with short normal-breathing periods (5 s) could prevent the induction of a significant HR change. We also hypothesize that this HR pattern could be derived from the task of walking while breath-holding, pushing the cardiovascular system to keep active. The previously mentioned HR decreases and increases could be clearly observed while holding the breath in a resting condition [37]. Further investigations should address whether the cardiac output produced during apnea depends on the interaction of the apnea itself (hypoxemia, hypercapnia, or exposure time) with the external stimulus (walking, running, cycling, at various volumes and intensities).

### 4.3. Mechanisms of Hypoalgesia

The mechanisms underlying apnea-induced hypoalgesia remain uncertain. However, from a broader perspective, one possible explanation is that apnea training might trigger mechanisms similar to those observed in conventional exercise. The involvement of several systems, including opioid, endocannabinoid, serotoninergic, immune and autonomous nervous system mechanisms, has been studied. Among these, serotoninergic, and certain immune markers have been shown to correlate with EIH [38]. It is also plausible that apnea training shares similar mechanisms with blood-flow restriction training [6].

Additionally, research has explored specific mechanisms of apnea–hypoalgesia. For example, a study of Reyes del Paso et al. [7] found that a breath-holding maneuver reduced pain perception during painful stimuli, suggesting the role of baroreflexes in this response [8].

In our current study, the findings provide preliminary indications of the potential hypoalgesic effect of apnea training. However, to confirm this effect and accurately determine its magnitude and variability, a larger sample size is required. Subjacent mechanisms, such as hypoxemia, hypercapnia, HR, or perceived exertion, should be further examined. While HR remained relatively stable across groups, there was a notable decline in SpO_2_ in AG. Future studies should investigate the underlying mechanisms of this finding in greater depth, particularly the potential role of hypoxia-induced cardiovascular adaptations. Similar variables have been examined as mediators of the hypoalgesic effect in other exercise modalities. For instance, in aerobic exercise, both intensity and duration modified EIH [39,40,41]. Naugle et al. [39] found a dose–response effect between exercise intensity and pain perception, with high-intensity exercise showed a greater effect compared to moderate-intensity exercise.

It should be noted that current evidence supports that EIH can be achieved through a wide variety of exercise modalities, likely due to the activation of overlapping action mechanisms [42]. Apnea-induced hypoalgesia may share similar mechanisms with conventional aerobic exercise, blood-flow restriction training, and breath-holding maneuvers.

### 4.4. Limitations

One limitation of this study is the small sample size, which reduces the statistical power and may explain why some potentially meaningful differences were not significant. The exploratory nature of this trial was intended to assess the feasibility of apnea interventions for pain sensitivity, but a larger sample size is necessary to confirm these effects and provide more robust conclusions.

Another limitation concerns the homogeneity of the sample, which consisted of healthy young adults of 19–29 years. This restricts the generalizability of our findings to other populations, particularly middle-aged, older adults or clinical populations. Although early-stage research typically focuses on young, healthy individuals to assess safety and physiological responses, future studies should investigate apnea interventions in clinical populations. This would allow researchers to evaluate the intervention’s effectiveness in clinical settings, including direct pain measures like pain intensity, or pain disability.

Additionally, the lack of long-term follow-up analyses means that we cannot assess whether the hypoalgesic effects of apnea training are sustained over time.

### 4.5. Future Perspectives

To our knowledge, this trial is the first to explore the potential hypoalgesic effects of apnea training. Apnea offers distinct advantages as individuals may achieve greater physiological stress with shorter intervention durations, making it easier to achieve higher exercise intensity. However, further research in healthy participants is needed to explore various apnea modalities. Future studies should investigate the combination of different lung volumes, session durations, apnea densities (continuous–intermittent), and external loads (rest–active). Additionally, monitoring rated perceived exertion, hypoxemia, hypercapnia, and HR should be examined in more detail. Specific biomarkers, such as blood lactate, cortisol levels or capnography, may also provide valuable insights in future studies.

Additionally, research should extend the inclusion of other quantitative sensory measures, such as thermal pain thresholds, temporal summation, and pain tolerance measures.

## 5. Conclusions

The current results suggest a trend, though not statistically significant, toward an improvement in the PPT in favor of the apnea training compared to normal breathing. No significant changes occurred in CPM response. However, larger sample sizes are needed to confirm these findings conclusively. Apnea intervention produced a significant reduction in SpO_2_, with a notable correlation between the reached desaturation and the PPT response. Additionally, basal CPM correlated with PPT changes in apnea training. Future research should aim to clarify these relationships and determine the underlying mechanisms behind these findings.

## Figures and Tables

**Figure 1 sports-12-00294-f001:**
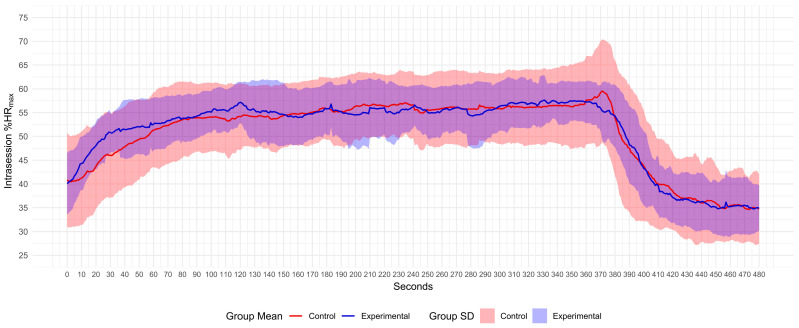
Mean %HR_max_ and SD response during intervention (0–360 s) and rest period (361–480 s). %HRmax, percentage of theoretical maximum heart rate.

**Figure 2 sports-12-00294-f002:**
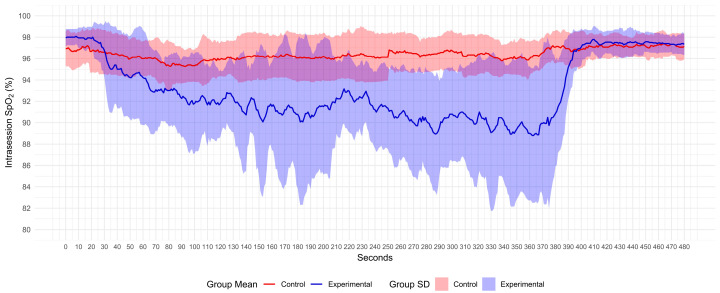
Mean SpO_2_ and SD response during intervention (0–360 s) and rest period (361–480 s). SpO_2_, oxygen saturation.

**Figure 3 sports-12-00294-f003:**
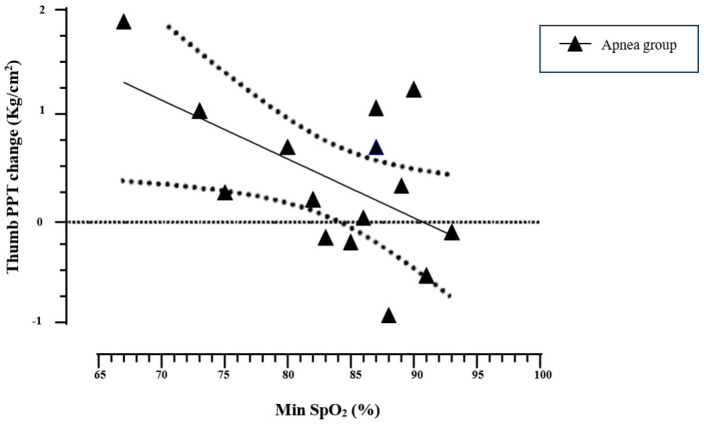
Pearson’s correlation of minimum SpO_2_ values and thumb PPT change in the apnea group. The linear regression line (solid line) and its 95% CI (dotted line) are presented in the plot. Min SpO_2_, minimum oxygen saturation value reached during apnea training.

**Table 1 sports-12-00294-t001:** Sociodemographic and baseline clinical data of participants.

	AG (n = 15)	CG (n = 15)	Pair-Wise Comparisons (*p*-Value)
Age (years)	23.93 ± 2.939	23.13 ± 1.99	0.39 ^‡^
Gender Male/Female (n)	10/5	10/5	1.00 ^$^
BMI (kg/m^2^)	22.89 ± 2.39	22.94 ± 2.16	0.95 ^‡^
GPAQ (METs)	4580 ± 3569.93	4700 ± 4349.04	0.89 ^§^
PSS-14	17.60 ± 6.46	17.67 ± 6.75	0.97 ^‡^
PSQI	5.27 ± 2.89	5.93 ± 2.34	0.30 ^§^
SpO_2_ (%)	98.20 ± 1.01	97.20 ± 1.42	0.035 ^‡,^*
%HR_max_ at rest	32.01 ± 4.56	32.97 ± 7.53	0.68 ^‡^
PPT Thumb (kg/cm^2^)	3.66 ± 0.72	3.5 ± 1.42	0.29 ^‡^
PPT Thumb cond. (kg/cm^2^)	4.40 ± 0.96	3.86 ± 1.33	0.12 ^‡^
PPT Tibialis (kg/cm^2^)	4.41 ± 1.65	4.22 ± 2.02	0.30 ^‡^

Mean ± SD were presented in the table. ^‡^, Student’s *t*-test was applied; ^§^, Mann–Whitney U test was applied; ^$^, chi-squared test was applied; *, *p* < 0.05. %HR_max_, percentage of theoretical maximum heart rate; AG, apnea group; CG, control group; BMI, body mass index; CPM, conditioned pain modulation; GPAQ, Global Physical Activity Questionnaire; METs, metabolic equivalent of task; PPT, pressure pain threshold; PSQI, Pittsburgh Sleep Quality Index; PSS-14, Perceived Stress Scale; SpO_2_, oxygen saturation.

**Table 2 sports-12-00294-t002:** Mixed ANOVAs of PPT and CPM responses.

Measures	Groups	Time-Point				Mixed ANOVA (F, *p*-Value; η_p_^2^)		
		Pre	CPM Pre	Post	CPM Post	Time	Group	Time × Group
PPT Thumb	AG	3.66 ± 0.72	-	4.06 ± 1.11	-	0.73, 0.39; 0.026	1.13, 0.29; 0.039	2.36, 0.13; 0.078
	CG	3.50 ± 1.42	-	3.38 ± 1.29	-			
PPT Tibialis	AG	4.41 ± 1.65	-	4.82 ± 1.87	-	0.64, 0.42; 0.023	0.46, 0.50; 0.016	2.17, 0.15; 0.072
	CG	4.22 ± 2.02	-	4.10 ± 1.99	-			
CPM and PPT Thumb	AG	3.66 ± 0.72	4.40 ± 0.96	4.06 ± 1.11	4.31 ± 0.83	4.70, **0.004 ***; 0.144	1.80, 0.19; 0.06	1.04, 0.37; 0.036
	CG	3.50 ± 1.42	3.86 ± 1.33	3.38 ± 1.29	3.68 ± 1.36			

Mean ± SD were presented in the table. *, *p* < 0.05. AG, apnea group; CG, control group; PPT, pressure pain threshold; CPM, conditioned pain modulation.

**Table 3 sports-12-00294-t003:** HR, SpO_2_, and summary data.

	AG (n = 15)	CG (n = 15)	Pair-Wise Comparisons (*p*-Value)
HR data			
Time in %HR_max_ Zone 1 (s)	215.20 ± 118.74	227.67 ± 121.95	0.78 ^‡^
Time in %HR_max_ Zone 2 (s)	114.67 ± 113.47	80.60 ± 96.99	0.45 ^§^
Time in %HR_max_ Zone 3 (s)	8.00 ± 25.75	45.73 ± 95.12	0.20 ^§^
Time in %HR_max_ Zone 4 (s)	0	0	-
Time in %HR_max_ Zone 5 (s)	0	0	-
SpO_2_ data			
Time in Normoxemia Zone (s)	119.53 ± 80.61	276.67 ± 122.30	**0.002** **^§,^** *****
Time in Mild Hypoxemia Zone (s)	65.53 ± 48.04	71.00 ± 120.13	0.16 ^§^
Time in Moderate Hypoxemia Zone (s)	112.67 ± 97.00	10.80 ± 33.00	**<0.001** **^§,^** *****
Time in Severe Hypoxemia Zone (s)	49.07 ± 68.84	0	**<0.001** **^§,^** *****

Mean ± SD were presented in the table. ^‡^, Student’s *t*-test was applied; ^§^, Mann–Whitney U test was applied; *, *p* < 0.05. %HR_max_, percentage of theoretical maximum heart rate; AG, apnea group; CG, control group; HR, heart rate; SpO_2_, oxygen saturation; Min, minimum.

## Data Availability

The data that support the findings of this study are available upon request to the corresponding author.

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
