# Peer review of "Effects of Apnea-Induced Hypoxia on Hypoalgesia in Healthy Subjects"

_sports, 2024, doi:10.3390/sports12110294_

Round 1
Reviewer 1 Report
Comments and Suggestions for Authors
The abstract provides a general overview of the paper's contents, but it could be more concise. The discussion of results is somewhat vague, particularly with regard to the statement "There were no changes in hypoalgesia between repeated apneas and aerobic exercise." Alternatively, the text should explicitly state that no statistically significant results were observed, while placing greater emphasis on the correlation between SpO2 and PPT response.
- Introduction: The introduction is comprehensive but could be shortened to focus more on apnea-induced hypoalgesia, thereby allowing the reader to gain a more concise understanding of the subject matter. For example, the discussion on blood flow restriction training (BFR) should be expanded to include the following:
- Apnea Protocol: The rationale behind the apnea intervention should be more clearly articulated. The rationale behind the alternating pattern of five seconds of breathing and ten seconds of low-volume apnea should be elucidated. Was this approach based on previous literature? In the absence of such justification, a rationale for this specific protocol should be provided.
- Baseline SpO2 Differences: The notable disparity in baseline SpO2 (though not clinically significant) between the groups merits further discussion. Even minor fluctuations in oxygen saturation levels may impact the outcomes of a study examining hypoxemia-induced hypoalgesia. Please elucidate how this might impact the results.
The results section asserts that no significant differences were identified between the groups. However, it also indicates that PPT responses were linked to basal CPM and hypoxemia in the apnea group. This finding is of significant importance and should be given greater emphasis. It would be beneficial for the manuscript to clarify whether the non-significant differences in PPTs between groups indicate that the apnea protocol was ineffective or whether this could be attributed to methodological factors, such as sample size or duration of apnea.
It is noteworthy that while heart rate remained relatively consistent across groups, there was a notable decline in oxygen saturation in the apnea group. The authors should investigate the underlying mechanisms of this finding in greater depth, with particular consideration of the potential role of hypoxia-induced cardiovascular adaptations.
- Statistical Analysis: In order to facilitate a more comprehensive understanding of the magnitude of the observed effects, it would be beneficial to include effect sizes in addition to p-values. Furthermore, although the use of a mixed ANOVA is appropriate, the manuscript would be enhanced by a more detailed explanation of the interaction effects and post-hoc tests.
- Mechanistic Insights: The discussion touches on potential mechanisms (e.g., hypoxemia, hypercapnia), but a more in-depth exploration of why apnea training might fail to produce significant hypoalgesic effects compared to aerobic exercise would be beneficial. It would be beneficial to ascertain whether specific physiological pathways, such as the opioid or endocannabinoid system, are activated during apnea.
- Comparison with Existing Literature: The authors acknowledge the dearth of prior studies on apnea-induced hypoalgesia; however, they should provide a more comprehensive comparison to other modalities, such as BFR or traditional aerobic exercise. What mechanisms might differentiate apnea from these other forms of exercise in terms of pain modulation?
The manuscript proposes future research, but the suggestions could be more specific. For example, the authors could provide more detailed suggestions for modifications to the apnea protocol, such as varying the duration or intensity of the apnea, or identify specific biomarkers that should be measured in future studies, such as blood lactate or cortisol levels.
Figures and Tables: The figures, particularly Figures 1 and 2, should be more detailed and informative. For example, providing more detailed labeling of the axes or including a legend that explains key variables could enhance comprehension.
Author Response
The abstract provides a general overview of the paper's contents, but it could be more concise. The discussion of results is somewhat vague, particularly with regard to the statement "There were no changes in hypoalgesia between repeated apneas and aerobic exercise." Alternatively, the text should explicitly state that no statistically significant results were observed, while placing greater emphasis on the correlation between SpO2 and PPT response.
Modifications have been conducted in the “Abstract” section, with changes concerning the writing of results and conclusion sections to enhance the comprehensiveness and suitability of the abstract.
Introduction: The introduction is comprehensive but could be shortened to focus more on apnea-induced hypoalgesia, thereby allowing the reader to gain a more concise understanding of the subject matter. For example, the discussion on blood flow restriction training (BFR) should be expanded to include the following:
Changes have been conducted in the “Introduction” section for further expose the current literature surrounding apnea-induced hypoalgesia and the hypothesised mechanisms behind this.
Apnea Protocol: The rationale behind the apnea intervention should be more clearly articulated. The rationale behind the alternating pattern of five seconds of breathing and ten seconds of low-volume apnea should be elucidated. Was this approach based on previous literature? In the absence of such justification, a rationale for this specific protocol should be provided.
Thank you for the suggestion of changes. As described previously in the literature, repeated apnea bouts have been confirmed to induce a reduction in oxygen saturation, a protocol with the potential to trigger the physiological changes related to apnea and therefore, with the potential to induce the hypoalgesic effect. Changes have been conducted in the “Interventions” (lines 93-100) section for further expose the current literature
Baseline SpO2 Differences: The notable disparity in baseline SpO2 (though not clinically significant) between the groups merits further discussion. Even minor fluctuations in oxygen saturation levels may impact the outcomes of a study examining hypoxemia-induced hypoalgesia. Please elucidate how this might impact the results.
Thank you for your valuable comment regarding the baseline SpO2 differences. The observed 1% difference in baseline SpO2 values between the groups, while statistically significant (p=0.035), was deemed clinically irrelevant due to all participants having SpO2 levels ≥95%. This minor fluctuation likely reflects normal variability between individuals rather than a meaningful deviation that could influence the study outcomes.
Given that the purpose of the study was to explore apnea-induced hypoalgesia through controlled apnea protocols, the critical factor is the reduction in SpO2 levels during the intervention, not the baseline levels. The baseline difference of 1% is unlikely to have a significant impact on the capacity of the individuals to experience hypoxemia or on the overall response to the intervention. Additionally, previous literature supports that short-term fluctuations within the normal range are not associated with significant physiological effects on oxygen-dependent processes.
We acknowledge the point raised and modifications specifying this concern will be placed within the manuscript, see “Participants” (lines 175-176) section, so the reader receives more detailed information about the impact of this difference.
The results section asserts that no significant differences were identified between the groups. However, it also indicates that PPT responses were linked to basal CPM and hypoxemia in the apnea group. This finding is of significant importance and should be given greater emphasis. It would be beneficial for the manuscript to clarify whether the non-significant differences in PPTs between groups indicate that the apnea protocol was ineffective or whether this could be attributed to methodological factors, such as sample size or duration of apnea.
Changes along the “Discussion” (lines 233-237, and 241-243) section have been made to better provide a more precise framework concerning the study results.
It is noteworthy that while heart rate remained relatively consistent across groups, there was a notable decline in oxygen saturation in the apnea group. The authors should investigate the underlying mechanisms of this finding in greater depth, with particular consideration of the potential role of hypoxia-induced cardiovascular adaptations.
Thank you for the comments. These concerns have been addressed in the “Discussion” (lines 246-250) section.
Statistical Analysis: In order to facilitate a more comprehensive understanding of the magnitude of the observed effects, it would be beneficial to include effect sizes in addition to p-values. Furthermore, although the use of a mixed ANOVA is appropriate, the manuscript would be enhanced by a more detailed explanation of the interaction effects and post-hoc tests.
Thank you for the feedback. The F-statistic, with its associated p-value, and partial-eta-squared as measures of fit of the ANOVA model were already present in the manuscript (Table 2). However, as the reviewer correctly requested, the addition of further effect sizes, for post-hoc analysis were additionally provided, including Mean Difference, standard error, t-value, and p-value of post-intervention differences between groups. See “Abstract”, “Statistical Analysis” (lines 163-164), “3.2 Pressure Pain Threshold” (lines 187-189) and “Discussion” (lines 233-237) sections.
Mechanistic Insights: The discussion touches on potential mechanisms (e.g., hypoxemia, hypercapnia), but a more in-depth exploration of why apnea training might fail to produce significant hypoalgesic effects compared to aerobic exercise would be beneficial. It would be beneficial to ascertain whether specific physiological pathways, such as the opioid or endocannabinoid system, are activated during apnea.
Thank you for the comment. Changes have been conducted in the “Discussion” (lines 273-279) section further providing possible action mechanisms behind apnea training.
Comparison with Existing Literature: The authors acknowledge the dearth of prior studies on apnea-induced hypoalgesia; however, they should provide a more comprehensive comparison to other modalities, such as BFR or traditional aerobic exercise. What mechanisms might differentiate apnea from these other forms of exercise in terms of pain modulation?
Changes have been conducted within the “Discussion” (lines 292-296) section, further addressing the suggestions proposed by the reviewer,
The manuscript proposes future research, but the suggestions could be more specific. For example, the authors could provide more detailed suggestions for modifications to the apnea protocol, such as varying the duration or intensity of the apnea, or identify specific biomarkers that should be measured in future studies, such as blood lactate or cortisol levels.
Modifications concerning reviewer’s suggestion have been addressed in the “Discussion” section (lines 302-304 and 305-306).
Figures and Tables: The figures, particularly Figures 1 and 2, should be more detailed and informative. For example, providing more detailed labeling of the axes or including a legend that explains key variables could enhance comprehension.
Thank you for the suggestions. Figures have been changed, including more detailed labelling, along with providing Mean and SD as dispersion parameters for detailing information concerning Intrasession values of HRmax(%) and SpO2(%) values. New Figures have been uploaded as Supplementary Material for them to be attached within the manuscript.
Reviewer 2 Report
Comments and Suggestions for Authors
I would like to acknowledge the efforts of the authors in implementing the project and writing this article “Effects of Apnea-Induced Hypoxia on Hypoalgesia in Healthy Subjects”.
This article brings interesting information about hypoalgesia and the possibility of influencing it through apnea..
The aim of this study is to explore the effect of apnea training on hypoalgesia; assess the correlation between conditioned pain modulation (CPM) response and apnea-induced hypoalgesia; and examine the association between hypoalgesia and rated perceived exertion (RPE), hypoxemia, and heart rate (HR) during apnea.
I have the following comments and questions:
What sample of the population were the participants selected from? How were they approached?
How was the number of participants determined?
Was the Power Analysis performed?
Line 137: I understand the use of this formula, but it's just a guess…
There were not too many positive results in the study. It might be worth trying to focus on the SpO2 values after the end of the intervention.
Author Response
What sample of the population were the participants selected from? How were they approached?
This information has been further indicated in “Participants” (lines 65-67) section.
How was the number of participants determined?
Was the Power Analysis performed?
The exploratory objective of the study implied the recruitment of a small sample to check the feasibility of the apnea manoeuvre in terms of adverse event responses, and the amount of hypoalgesic effect and its relevance. Therefore, before conducting a large clinical trial, authors decided to check this in a sample of 30 participants. This has been explained in “Participants” (lines 76-77) section.
Line 137: I understand the use of this formula, but it's just a guess…
There are several proposals in the literature for estimating maximal heart rate. One of the first proposals were the formula 220 – Age (Fox SM 3rd, Naughton JP. Physical activity and the prevention of coronary heart disease. Prev Med. 1972 Mar;1(1):92-120. doi: 10.1016/0091-7435(72)90079-5. PMID: 5069016). However, more recent proposals have emerged based on proper regression analyses, such as the one presented in the manuscript by Tanaka et al., (2001), or others such as Gulati et al., (2010). There is no such a consensus on whether choosing one formula or other. However, the most plausible and suitability of choosing a specific formula would be based on the similarity across the population tested and the present in our study.
This is the case for the formula of Tanaka et al., (2001), as they tested this formula in healthy women and men.
There were not too many positive results in the study. It might be worth trying to focus on the SpO2 values after the end of the intervention.
In complete agreement. We have reformulated the “Abstract” and “Conclusions” (lines 310-313) section according to your suggestions.
Round 2
Reviewer 1 Report
Comments and Suggestions for Authors
sorry but even if the topics is interesting it is still insufficient for publication :
because of that following 10 points:
- Generalizability:
- The study was conducted on healthy young adults aged 18-30.
This limits the generalizability of the findings to other age groups, individuals with chronic pain, or those with different health conditions.
- Control Group Protocol:
- The control group performed normal breathing while walking, which may not be an adequate control for the apnea intervention. A more rigorous control could involve a sham intervention that mimics the apnea protocol without inducing hypoxia.
- Measurement of Hypoalgesia:
- The study used Pressure Pain Threshold (PPT) and Conditioned Pain Modulation (CPM) as measures of hypoalgesia.
While these are valid measures, the study could benefit from including additional pain sensitivity measures, such as thermal or electrical pain thresholds, to provide a more comprehensive assessment.
- Duration and Intensity of Apnea:
- The apnea protocol involved 10-second breath-holds, which may be too short to induce significant physiological changes.
Longer apnea durations or different apnea patterns could potentially yield more pronounced effects.
- Lack of Long-term Follow-up:
- The study only assessed immediate post-intervention effects. Long-term follow-up would be valuable to determine if the hypoalgesic effects of apnea training are sustained over time.
- Potential Confounding Variables:
- Although the study controlled for some variables like physical activity level and sleep quality, other potential confounders such as diet, hydration status, and psychological factors were not controlled or reported.
- Mechanistic Insights:
- The study suggests potential mechanisms like hypoxemia and heart rate changes but does not provide direct evidence for these mechanisms.
Future studies should include biomarkers (e.g., blood lactate, cortisol) to better understand the underlying mechanisms.
- Blinding and Randomization:
- While the evaluator was blinded to group allocation, it is not clear if participants were blinded to the intervention, which could introduce bias. Additionally, the randomization process should be described in more detail to ensure it was adequately performed.
- Ethical Considerations:
- The study mentions ethical approval and informed consent, which is good practice.
​ However, the potential risks associated with apnea training, such as severe hypoxemia, should be more thoroughly discussed.
- Clarity and Readability:
- The article contains some typographical errors and formatting issues (e.g., table headings and text alignment) that should be corrected for better readability.
Overall, while the study provides interesting preliminary findings on the potential hypoalgesic effects of apnea training, it would benefit from addressing these critiques to strengthen the validity and reliability of the results.
Author Response
sorry but even if the topics is interesting it is still insufficient for publication:
because of that following 10 points:
We would like to thank you for taking the time to review our work. We appreciate and consider each of your suggestions. In our response we will try to resolve them, if possible, or include them as limitations of the study.
Generalizability:
The study was conducted on healthy young adults aged 18-30.
This limits the generalizability of the findings to other age groups, individuals with chronic pain, or those with different health conditions.
Response: We acknowledge that focusing on healthy young adults limits generalizability. However, this is a standard approach in early-stage research to evaluate the feasibility of the intervention in terms of safety, physiological responses, and participant tolerance. The study aimed to explore potential effects on pain-related measures in a controlled environment. Given that this population does not experience chronic pain, direct pain measures (intensity, frequency, disability) could not be assessed. Therefore, we focused on sensory-discriminative domains such as pain sensitivity thresholds and pain tolerance. This is a necessary first step in opening a new research line. Future studies should extend this research to other populations, including those with chronic pain, once the intervention's effectiveness and safety are further confirmed.
These concerns have been integrated in the “Limitations” section of the manuscript’s discussion.
Control Group Protocol:
The control group performed normal breathing while walking, which may not be an adequate control for the apnea intervention. A more rigorous control could involve a sham intervention that mimics the apnea protocol without inducing hypoxia.
Response: Currently, there are no well-established sham or placebo controls for exercise-related interventions or breath-holding maneuvers. Our comparison between breath-holding and normal breathing was chosen for its ecological validity and the practical relevance of breath-holding as a simple, accessible technique. We considered using an oxygen mask to produce reduced oxygen levels for the experimental group, and a simulated oxygen reduction in the group, but this would have neglected other physiological mechanisms related to apnea (e.g., hypercapnia, baroreceptor activation) that may contribute to hypoalgesia. Other studies have employed other types of placebo as there is currently no sham/placebo that matches exercise interventions. This is the case for the use of placebo TENS, or placebo ultrasound, or placebo oral pills, compared to exercise (Miller et al., 2022). However, clear limitations from the unmatched active-placebo interventions rise concerns from the validity of testing placebo in exercise trials.
As this was an exploratory study, we deemed a normal-breathing control appropriate. Future studies could incorporate more complex control designs, including sham protocols, to isolate specific effects.
Reference: Miller CT, Owen PJ, Than CA, Ball J, Sadler K, Piedimonte A, Benedetti F, Belavy DL. Attempting to Separate Placebo Effects from Exercise in Chronic Pain: A Systematic Review and Meta-analysis. Sports Med. 2022 Apr;52(4):789-816. doi: 10.1007/s40279-021-01526-6
Measurement of Hypoalgesia:
The study used Pressure Pain Threshold (PPT) and Conditioned Pain Modulation (CPM) as measures of hypoalgesia.
While these are valid measures, the study could benefit from including additional pain sensitivity measures, such as thermal or electrical pain thresholds, to provide a more comprehensive assessment.
Response: PPT and CPM were chosen for their reliability and feasibility within our research setting. We did not include additional pain sensitivity measures to avoid potential interference between different modalities and to ensure timely post-intervention assessment. We recognize that incorporating measures like thermal or electrical pain thresholds could provide a more comprehensive evaluation of hypoalgesia, and we will consider this in future studies to broaden the scope of pain sensitivity assessment.
This has been included in the new section “4.5. Future perspectives”.
Duration and Intensity of Apnea:
The apnea protocol involved 10-second breath-holds, which may be too short to induce significant physiological changes.
Longer apnea durations or different apnea patterns could potentially yield more pronounced effects.
Response: While the breath-holds were relatively short, they still induced measurable physiological changes, as seen in the oxygen saturation response (see Figure 2 and Table 3). This intermittent pattern produces incomplete oxygen saturation recoveries between apneas, and produces the observable desaturation pattern. Longer apnea periods would require additionally longer recovery periods with the potential of not inducing the sufficient physiological stress to induce the aimed hypoalgesic response. In addition, the apnea bouts’ duration was selected to ensure participant safety and compliance, particularly as they were untrained in apnea. We agree that longer breath-holds could amplify the effects, however it is possible that they are not feasible for all participants. Future studies should explore variations in apnea duration and patterns to assess their impact more fully.
Lack of Long-term Follow-up:
The study only assessed immediate post-intervention effects. Long-term follow-up would be valuable to determine if the hypoalgesic effects of apnea training are sustained over time.
Response: We agree that long-term follow-up would provide valuable insights into the duration of the hypoalgesic effects. Due to logistical constraints, we focused on immediate effects in this study. Future research should include follow-up assessments to investigate whether these effects are sustained over time.
This concern has been included in a new section of limitations “4.4. Limitations”.
Potential Confounding Variables:
Although the study controlled for some variables like physical activity level and sleep quality, other potential confounders such as diet, hydration status, and psychological factors were not controlled or reported.
Response: We implemented strict inclusion criteria to minimize potential confounding variables. Additionally, we controlled for factors commonly associated with hypoalgesia, such as physical activity, sleep quality, and perceived stress (a type of psychological variable as suggested in the comment). In any case, the randomization process was implemented in order to similarly distribute potential confounding variables across groups. It should be noted that the randomization process was enhanced with the stratification by blocks based on sex and age groups.
Mechanistic Insights:
The study suggests potential mechanisms like hypoxemia and heart rate changes but does not provide direct evidence for these mechanisms.
Future studies should include biomarkers (e.g., blood lactate, cortisol) to better understand the underlying mechanisms.
Response: To focus on a mechanistic insight, it should be well-proven that the intervention really provides a valuable effect. The main purpose of the study was analysing the effectiveness, and feasibility of the intervention. In addition, we further analysed possible mechanistic processes potentially involved in this response, which were easy to measure with our available devices, which included HR, saturation. These concerns have been included in “4.3. Mechanisms of hypoalgesia” section.
Blinding and Randomization:
While the evaluator was blinded to group allocation, it is not clear if participants were blinded to the intervention, which could introduce bias. Additionally, the randomization process should be described in more detail to ensure it was adequately performed.
Response: Participants were not blinded to the intervention, as it is not possible in exercise trials. Therefore, they were not blinded to the assessment of PPTs or CPM. It should be noted that the procedure for assessing PPTs and CPM with algometry, involves intrinsically two sources of potential error: 1) the intrinsic error from self-reported measures, as the participant knew the received intervention. It is currently not possible to incorporate blinding methods to participants in exercise trials (indicated in the above-mentioned reply); and 2) the error provided by the assessor with the algometer, which was solved with the blinding.
The randomization process was carried out by an external researcher using GraphPad software, as stated in the manuscript, which ensures clarity and transparency in the randomization methodology.
Ethical Considerations:
The study mentions ethical approval and informed consent, which is good practice.
​However, the potential risks associated with apnea training, such as severe hypoxemia, should be more thoroughly discussed.
Response: The ethical committee thoroughly reviewed the potential risks associated with the apnea protocol, and no participants withdrew due to adverse effects. The intervention was well-tolerated, and we ensured close monitoring to mitigate the risks of hypoxemia. In any case, the informed consent stated that at the first sign of severe hypoxemia, such as ataxia or loss of motor control, the protocol would be stopped immediately.
This information has been included in “2.1. Participants” section.
Clarity and Readability:
The article contains some typographical errors and formatting issues (e.g., table headings and text alignment) that should be corrected for better readability.
Thank you for notifying it, changes in formatting have been conducted in Table 1, Table 2, and Table 3. Additionally, the Figures of HR and SpO2 previously provided as supplementary material were now incorporated within the manuscript.
Overall, while the study provides interesting preliminary findings on the potential hypoalgesic effects of apnea training, it would benefit from addressing these critiques to strengthen the validity and reliability of the results.
Response: We will carefully review the manuscript to correct any typographical errors and formatting inconsistencies to enhance clarity and readability. We have additionally stratified the discussion into 5 sections to enhance the comprehension of the information. Among these sections “Main findings”, “Cardiovascular and respiratory responses”, “Mechanisms of hypoalgesia”, “Limitations”, “Future perspectives”, and “Conclusions”, we aim to provide valuable insights which address both the presented information and the comments suggested by the reviewer.
Thank you for your thorough review and valuable suggestions. We believe these revisions will significantly strengthen the quality of manuscript along with its new insights to the literature.
Reviewer 2 Report
Comments and Suggestions for Authors
My comments have been implemented.
Author Response
My comments have been implemented.
Dear Reviewer,
Thank you very much for your valuable feedback on the first revision of my manuscript. I’m glad the changes have addressed your comments. Your input has been greatly appreciated.
Best regards,
Round 3
Reviewer 1 Report
Comments and Suggestions for Authors
you have improved the manuscript.